# Reading the MAP: A Pracademic Perspective on the Current State of Play of the Multi-Action Plan Model with Regard to Transitions between Mental States

**DOI:** 10.3390/ijerph192315520

**Published:** 2022-11-23

**Authors:** Bernadette Kellermann, Alan MacPherson, Dave Collins, Maurizio Bertollo

**Affiliations:** 1Institute for Sport, Physical Education and Health Sciences, University of Edinburgh, St. Leonard’s Land, Holyrood Road, Edinburgh EH8 8AQ, UK; 2Grey Matters Performance Ltd., Stratford upon Avon CV37 9TQ, UK; 3Department of Medicine and Aging Sciences, University “G. d’Annunzio” of Chieti-Pescara, 66100 Chieti, Italy

**Keywords:** peak performance, coping, self-regulation, biopsychosocial, flow state, clutch

## Abstract

The Multi-Action Plan (MAP) presents as an action-focused, sport-specific, mixed methods intervention model. MAP research characterized four Performance Types (PTs). Each PT operates on an affective, cognitive, behavioral, and psychophysiological level—across performance contexts. In this narrative review, we present a synthesis of our current understanding of MAP research, coupled with offering applied implications and directions for future research. We make the case for investigating the timing of transitions between PTs as our primary area of interest in expanding the MAP framework on a conceptual and applied level. Regarding pre-transition cues, we offer ideas on examining socio-environmental precursors to performance, with the aim of expanding MAP from a psycho-bio (affective, cognitive, behavioral, and psychophysiological dimensions) to a biopsychosocial concept (affective, cognitive, behavioral, psychophysiological, and socio-environmental dimensions). Regarding post-transition, we propose that investigating short- and long-term effort and reward perception will yield valuable insights into athletes’ rationales behind the selection, operationalization, and experience of specific PTs. Finally, and from a pracademic perspective, we reflect critically on the achievements of MAP research thus far and provide specific directions for future research.

## 1. Introduction

Over the past few decades, multiple attempts were made to conceptualize peak and non-peak performance experiences within sport psychology [1]. Peak performance can be defined as a state of superior functioning resulting in optimal human performance [2]. Most peak performance research can be characterized as either in-depth descriptive studies (e.g., [3]) or were empirical in their orientation [4]. However, broadly, the aim of the peak/non-peak research paradigm is to characterize and determine what does and does not constitute a peak experience and its associated antecedents and consequences. In contrast to these dichotomous, peak or non-peak approaches sits the Multi-Action Plan (MAP) Model. MAP is an intervention model that can be utilized to investigate and improve human performance, using various research methodologies. To date, the body of research utilizing MAP investigated the execution of self-paced skills (e.g., [5]). Mechanistic approaches such as this can offer ideas on the ‘How?’ and ‘Why?’ of human behavior in complex, high-stakes, and dynamic performance environments. They are crucial aids that facilitate effective applied practice, and enable accurate case conceptualization, assisting both clients and practitioners (cf. [6]).

Introduced by Bortoli and colleagues [1], MAP provides a conceptual structure that outlines both peak and non-peak performance; perceived emotional states; the control a performer has over their performance situation; and potential coping strategies. Therefore, although providing a psychophysiological perspective, MAP is idiosyncratic, meaning that it can be applied accurately to an individual’s circumstances, as opposed to offering generic guidance.

Given that high-performance environments in sport are dynamic and unpredictable, the experiences of performers examined within the MAP framework are inherently complex, elusive to systematic study, and challenging to control [1]. Yet, to date, MAP was tested and corroborated in nine studies across multiple self-paced activities (shooting and dart throwing [1,5,7,8,9]), endurance sports (cycling [10,11]; running [12]), and in driving simulation [13].

MAP is steeped in the Multi-States Theory for Self-Regulation, which conceptualizes the idiosyncratic experience of and dynamic interaction between performer, task (process, outcome, and performance context), perceived resources, and emotion- and action-based self-regulation [14]. Furthermore, the MAP model presents a novel conceptualization of high-performance experiences, synthesizing existing research frameworks, including Individual Zones of Optimal Functioning—a framework profiling a performer’s idiosyncratic experience of arousal, pleasantness, and functionality of emotion [15,16]; the Mindfulness—Acceptance—Commitment approach—a framework promoting nonjudgmental acceptance of cognitions, emotions, and sensations as part of self-regulatory strategies [17]; the Identification—Control—Correction (ICC) program—an evidence-based psycho-pedagogical approach to optimize the coaching and performance of elite athletes [16]; and flow state—a harmonious, rewarding, but elusive peak experience of performance (e.g., [18]).

Importantly, however, MAP research does not “cherry pick” elements of these frameworks and models. Rather, MAP proposes it is possible that tenets of these theories can apply at the same time—albeit in an idiosyncratic manner [1]. Having taken this holistic approach, proponents of MAP added a mechanistic layer to existing work and were, therefore, able to provide answers as to why certain cognitive and/or emotional processes may occur prior and during performance. For example, MAP contributes to the “demystification” of the much-described flow state, offering insight into how to get oneself *into* flow by applying an emotion-focused strategy [1]. This contrasts with previous, purely descriptive research on *being in* flow.

### 1.1. Purpose and Research Aims of This Narrative Review

Reflecting this important distinction, and with respect to this body of work, we firstly present a synthesis of our current understanding of MAP research, as well as offer our ideas on how MAP research could be expanded in the future. Our focus will be on transitions between mental states; specifically, upregulation—transitioning from less effective to a more effective mental state—and downregulation, where athletes transition from a more effective to a less effective mental state. We recognize that a primary purpose of sport and performance psychology is the investigation and practical application of upregulation processes aiming to re-gain, optimize, and/or stabilize performance. However, to develop an accurate appreciation of peak performance experiences, careful note must also be made of the factors that contribute to downregulation. Consequently, in relation to the MAP framework, we conceptualize existing research that examines both up and down regulation, the primary aim being to expand upon this by developing an approach to investigating and enhancing human performance.

We will present our findings in the format of a narrative review, considering the number of published studies currently utilizing the MAP framework and respecting the idiographic approach taken. Specifically, for studies to be included in this review, the study protocol required to utilize the MAP framework to conceptualize research findings. A total of nine studies were identified [1,5,7,8,9,10,11,12,13], based on the author’s knowledge of the literature. This number was confirmed by extensive searches of several databases. As an idiosyncratic framework, MAP highlights individual performance, as opposed to generalisable findings [1]. Accordingly, we will present our findings in the format of a narrative review, respecting the nuance of such an individualised and complex approach. Before offering three areas on how MAP could be extended, we will present our understanding of the current state-of-play of the MAP literature, as well as any gaps in this body of research.

### 1.2. Reading the MAP—Characteristics Performance Types 1–4

MAP consists of four Performance Types (PTs): Types 1 and 2 signify optimal performance, whereas in Types 3 and 4, the athlete is performing sub-optimally [1]. In addition to quality of performance, PTs differ in respect to each other, and in terms of the level of control perceived/exerted by the athlete: Types 1 and 4 are characterized by automatic performance, whereas Types 2 and 3 are effortful for the athlete to maintain [1].

Type 1 performance is characterized by confident, optimal monitoring of task [1]. Physical and mental resources are optimally available to the athlete, resulting in both a positive challenge appraisal and positive emotional experience: movement execution appears functional, automatic, and consistent. In contrast with Type 1, Type 2 is characterized by an increased need for, and level of, internal control over the task at hand. Due to this increased exertion of effort, resources required to maintain focus and energy are recruited [1]. Consequently, the athlete experiences functional, but unpleasant emotions overall that result in a threat appraisal.

While Types 1 and 2 both produce optimal performance, Type 3 is characterized by suboptimal performance due to heightened focus on task-irrelevant components that inhibit automatic movements [1,19]. This overcompensation and resultant increase in attentional effort is commonly accompanied by unhelpful cognitions and negative-dysfunctional emotions that also result in a harm/negative appraisal by the performer [1]. Lastly, Type 4 performance is characterized by action withdrawal and a dysfunctional-pleasant emotional profile, resulting in ineffective resource recruitment (i.e., behavioral, cognitive, and emotional) for the task at hand [1]. While the athlete might prematurely appraise the benefit of task completion, ineffective movements appear to be lacking in control, resulting in suboptimal performance.

PTs also differ on the physiological and neural levels. To date, analyses of psychophysiological and neural markers complemented psychological and behavioral results within and across studies, such as skin conductance level [9] and skin temperature [13]; heart rate [9,10,11,12,13]; respiratory rate and postural adjustments [9,13]; VO_2_ [10,11,12], VCO_2_ [10,11], and blood lactate [12]; and neural markers and cortical activity [8,9,11]. For example, skin conductance levels of a shooter and a dart-thrower showed differences between effortful PTs (Types 2 and 3) and more automatic states (Types 1 and 4), which was largely reflected in respiratory and postural data [9]. In contrast, psychophysiological analyses conduced in drivers mostly differed between optimal (Types 1 and 2) and suboptimal states (Types 3 and 4). On a neural level, PTs are associated with different cortical patterns. Specifically, automatic states (Types 1 and 4) are typified by event-related synchronization, whereas effortful states (Types 2 and 3) are typified by event-related desynchronization [8,9]. Further, under high exertion, Type 3 is characterized by high EEG coherence compared with Types 1 and 2 [11].

### 1.3. Reading the MAP—Operationalizing Performance Types 1–4

Currently, studies utilizing the MAP framework investigate one PT per performance in shooting and dart-throwing [1,5,7,8,9], cycling [10,11], running [12], and driving [13]. Further, while most MAP studies examined all four PTs [1,5,7,9,12,13], two studies focused on Types 1–3 [10,11] and one study on Types 1 and 2 [8]. However, research on the phasing of performance indicated that one performance may consist of multiple phases and/or stages; for example, there may be sub-phases of increased pressure *during* one longer performance [20]. Thus, whilst MAP currently characterizes the operationalization of PTs, transition mechanisms that enable performers to self-, or even upregulate, are yet to be investigated.

In terms of self-regulated transitioning between, and attainment of, PTs, there is a clear distinction between the operation of action- and emotion-focused coping strategies [1]. For example, emotion-focused strategies are necessary to attain flow-like Type 1 performance experiences. In contrast, athletes may apply action-focused coping strategies to regain optimal Type 2 performance following a suboptimal performance episode [1]. In terms of how transition mechanisms are initiated, the Affect as Information Theory suggests that affect can act as meta-cognitive cue(s) for transitioning between distinct cognitive processing styles [21]. Due to individual differences in the way performers recruit psychological resources (e.g., focus) to control performance in PTs, it can be assumed that affect can cue the processes involved in transitioning between Types 1–4. However, as we elaborate in Section 2 of this manuscript, research is needed to examine these cues in isolation and combination.

### 1.4. Contextualizing Types 1 and 2 with Related Peak Performance Concepts

Before offering our ideas on how to expand the MAP model, we highlight commonalities and differences with related peak performance concepts, to better understand the specificity around the MAP typology. Characteristics of Type 1 performance are widely consistent with those associated with flow state, including complete absorption, automaticity of movements, and absence of irrelevant or unhelpful cognitions and emotions [22]. Expanding on Csikszentmihalyi’s work, Swann and colleagues suggested a framework describing “letting it happen” performance, which is akin to a flow-like experience [23]. There are clear and marked similarities between flow, “letting it happen” [23], and Type 1 performance that include minimally conscious monitoring of relevant task components, complete absorption in the activity, and positive affect.

In contrast, Type 2 is characterized by a threat appraisal, unpleasant, albeit functional emotions, and a desire to upregulate to Type 1 performance [1]. Importantly, however, performance output is, like Type 1—optimal. These characteristics are reflected in “making it happen” performance episodes—the “letting it happen” counterpart that outlines optimal, focused, but effortful performance [23]. However, “making it happen” is also characterized by a challenge appraisal, positive emotions, increased motivation, and flow-like changes in perception (e.g., regarding time [23]). These aspects present crucial differences to MAP’s Type 2. In short, while characteristics such as effective recruitment of resources and increased focus and awareness are shared between Type 2 performance and “making it happen”, they may be still distinct experiences.

As a concept applicable to both Types 1 and 2, clutch performance is defined as episodes of superior performance despite increased pressure during a performance event [20,24]. Specifically, flow-like experiences such as automaticity and the absence of negative thoughts correspond with characteristics associated with Type 1 performance [1,20]. However, deliberate focus, increased effort, heightened perceived control, and increased arousal, resemble the Type 2 performance attributes [1,20]. While a Type 2 experience shares facets with both clutch and “making it happen” [23], concept-specific differences remain with both the clutch concept, i.e., automaticity in clutch vs. consciously controlled movements in Type 2, and “making it happen” [23] or challenge appraisal in “making it happen” vs. threat appraisal in Type 2.

In conclusion, while it is relatively clear that, within MAP, Type 1 performance is associated with flow-like experiences, the mechanisms proposed as underpinning Type 2 are more complex. Indeed, and potentially confusing for application, it appears that both “making it happen” as well as the clutch concept share characteristics with both Type 1 and Type 2. This indicates that Type 2 may in fact be more complex regarding characteristics such as appraisal and emotions, but also more realistic and frequent, especially in comparison with Type 1 and flow-like experiences. However, accurate comparison is problematic given the relative paucity of data regarding Type 2-like performance experiences, in contrast with the comprehensive body of literature investigating Type 1/ low experiences.

### 1.5. Extending the MAP—Considering Important Next Steps

While MAP provides a valuable framework for conceptualizing peak performance, there are still several areas that require further investigation. Accordingly, we next overview the current understanding of MAP and highlight avenues for future research. Findings are presented in the format of a narrative review, summarizing the current state-of-play of research conducted within MAP. First, we argue that a better understanding of the phasing/timing of transitions between mental states during performance would greatly expand the MAP framework on a conceptual and applied level. Second, we offer our ideas on the role of socio-environmental precursors acting as cues for such transition processes. Third, we propose clarification of the role of effort and reward perception post-transition, particularly in relation to micro and macro timeframes.

## 2. Transitions between Mental States: Performance as a Multi-Phase Process

### 2.1. Current Understanding

Currently, studies within the MAP framework are conducted on the apparent assumption that an athlete engages in one PT per performance. This point is reinforced as participants included in experimental studies are assigned exclusively to an experimental condition that corresponds to one PT, which is subsequently confirmed by the administration of post-experimental manipulation checks [10,11,12]. While this methodological decision is well-justified in the study context and supports the purpose of examining PT characteristics for the sake of categorisation, we argue that performers may experience multiple PTs during one performance—a consideration currently not reflected in the MAP literature.

Within the MAP framework and representing (in our opinion) a particular strength of the model, each PT is characterized by differences in appraisal, emotional experience, resource recruitment, attentional focus, level of control, and mental strategy, to name only some of the cognitive and emotional components that feature [1]. While these characteristics build the foundation to multi-dimensional PTs, to-date, studies adopting the MAP framework categorized MAP PTs according to research question(s), methodology, and participant expertise. Most commonly, PTs were categorized using median split techniques, involving objective performance scores and participants’ ratings of perceived control [4,7,8,9,13]. This method was applied with expert shooters and drivers. In contrast, (mostly) non-elite endurance athletes were *assigned* to a PT—specific experimental conditions, based on corresponding categorization variables [10,11,12]. Once again, manipulation checks were administered to ensure the correct assignment of participants. Unfortunately, however, as a result, we suggest that some “explicative power” regarding the idiosyncratic experience of PTs may be lost, particularly against the previously identified strengths of an individualized and idiographic approach.

In summary, given that negotiating unexpected stressors is a major contributory factor to complexity of performance, it is likely that a performer may have to operationalize and apply more than one PT when performing. Thus, while MAP research yielded highly informative data thus far, incorporating the impact of *transitions* between PTs during performance could prove an impactful new area of MAP research.

### 2.2. Making the Case for Performance as a Multi-Phase Process

In proposing this direction, we suggest that a rationale for doing so is already provided by research into performance states. For example, qualitative and meta-analytic evidence indicate that athletes may perceive clutch performance on an episodic level (micro timeframe), *as well as* appraising entire events as a clutch performance (meso timeframe) [20,25]. Specifically, the timing of when athletes appraise performance as clutch or as a flow state may differ depending on their perceived ability to cope. This will impact upon their ability to enter, exit, and maintain a particular PT or combination thereof [26]. To the best of our knowledge, clutch performance was not investigated using a MAP framework. Moreover, evidence referred to previously [20,25] suggests that clutch performance can be episodic in nature, thus increasing the likelihood that performance comprises multiple phases. Importantly, this facet appears to vary based, at least in part, on athlete perception.

Notably, however, in-performance episodes were also studied at a neural level. A further example regarding flow state indicated that when participants encountered an unexpected stressor during tightrope performance, EEG patterns were distinctly different from patterns recorded during flow state [27]. We would suggest that a disruption of a Type 1 performance, viewed through the MAP paradigm, may have caused a transition to Type 2 or Type 3 *within the performance episode*. Findings indicate that disruption to Type 1 performance is a realistic scenario for performers and practitioners to navigate [27]. Therefore, coping and upregulation, in order to stabilize or and re-gain an optimal performance state are key psychological skill elements to maintain effective performance against variations in challenge.

### 2.3. Implications: Types 1–4 as Micro, Meso, and Macro Performance Episodes

Based on these observations, we suggest that Type 1–4 performances may all appear as episodes during one longer performance [20,25,27]. For example, a cyclist may experience five minutes of Type 1, followed by 20 min of Type 2 and 15 min of Type 3 during their 40-min ride. This would constitute three short (micro) performance episodes, across three PTs, during one longer (meso) performance event (e.g., a Tour event). However, their experience may differ over the course of a season (macro), considering that tasks and contexts will vary greatly. Consequently, it may be an oversimplification to assume that athletes operationalize only one PT per performance. This also has implications for research study design, as to-date MAP studies were “strengthened” by pre-assigning participants to PTs [10,11,12].

Of course, these possibilities do seem to have been considered. For example, first steps were made in hypothesizing the transition process from Type 3 to Type 2 as a multi-stage process involving mindful acceptance [1]. In-performance transitions are not empirically supported, however. Further research, perhaps in the form of qualitative evidence or mixed design research, is needed to investigate the intra-transition processes outlined. For example, stimulated recall through event-focused interviews, conducted soon after a performance event, may give valuable insight into how athletes characterize their performance in terms of PT phasing and strategies that were utilized in a specific performance context [26]. A second method could be think-aloud protocols to gain insight into the cognitions employed during transitions, concurrently or retrospectively; however, think aloud, by design, “only” report thoughts but lack reflective explanations as to the intention behind why they occurred [28]. This element would be an important addition to the existing protocol.

## 3. Pre-Transition: Socio-Environmental Precursors of Performance

To date, MAP adopted a psycho-bio perspective to conceptualize optimal and suboptimal performance. Within any given performance situation, however, performers are required to evaluate the performance task through their internal experience *and* the social context in which performance occurs. Notably, this psychosocial facet has yet to be investigated in relation to MAP. For example, self-efficacy and the performer’s perception of control are important precursors in the interaction between self and environment in relation to their performance. Indeed, we would suggest that they act as crucial beliefs, underpinning performer decision-making and drive selection of and switching between PTs.

### 3.1. Self-Efficacy (Yet to Be Examined in MAP)

Defined as the belief in the ability to execute a specific task effectively [29,30,31], self-efficacy was extensively researched in the sporting context over the past few decades. Within MAP research, however, self-efficacy was not examined. Of course, it could be hypothesized that performers in Type 1 or 2 states possess high levels of self-efficacy and are, therefore, more likely to perform optimally. For example, in a Type 1 state, the performer’s skills optimally match task demands [32]. Similarly, evidence suggests that self-efficacy increases when perceived task difficulty decreases [33]. Consequently, it can be speculated that an athlete’s self-efficacy level in Type 1 might be optimally calibrated in relation to perceived task difficulty. In contrast, performers in Types 3 and 4 states might experience higher levels of perceived task difficulty due to increased task demands and insufficient coping resources. Therefore, athletes operationalizing Types 3 and 4 may have lower self-efficacy beliefs. Considering the impact of performance accomplishments and self-efficacy, this may perpetuate the suboptimal state and lead to further suboptimal performance.

This dynamic posits a link between performers’ self-efficacy beliefs and the generation and maintenance of psychological momentum. While psychological momentum can be both positively geared toward optimal performance, but also sometimes, exacerbate negative suboptimal performance, self-efficacy was found to increase positive, psychological momentum [34]. Future research is needed to gain insight into the role of self-efficacy in the selection of and/or transitions between PTs, specifically in response to previous performance.

Of course, both macro and micro dynamics require investigation; as an example for a possible macro timeframe, a performance by an athlete that took place earlier in the season could be interpreted by them as Type 2. In order to better understand the link between PT appraisal and momentum, it would be instructive to determine how the perception of this experience then influences the performer’s self-efficacy profile for the remainder of the season, considering that task difficulty will vary during the course of a season [33]. In short, what factors influence psychological momentum, be it towards optimizing or stabilizing performance? What PTs do the athlete engage in, or try to employ, and how can these be characterized? Moreover, future research is required to better understand how self-efficacy impacts acutely, on a micro timeframe. Reflecting our earlier argument on conceptualizing performance as a multi-phase process, across the course of one performance, an athlete might start a performance in Type 3 PT, with lower self-efficacy beliefs, but utilize the necessary psycho-social resources to rally and improve. However, are such changes influenced a priori by self-efficacy or does this improve *after* the change is accomplished? Future research is required to investigate the coping strategies required for upregulation towards optimal performance and the role that self-efficacy plays.

Consequently, as self-efficacy increases, it can be hypothesized that athletes are better able to regulate their affective experience [31,35]. In the context of the emotional profile of each PT, this could be of real interest: pleasant-functional (Type 1), unpleasant-functional (Type 2), unpleasant-dysfunctional (Type 3), and pleasant-dysfunctional (Type 4) [1]. While MAP proposes emotion- and action-focused self-regulation strategies for movement between PTs, future research is required to investigate the role of self-efficacy as a precursor to effectively employ such coping strategies. In addition, increased self-efficacy might indirectly lead to increased positive affect, through its role as a precursor to better performance [36]. For example, could vicarious experiences act as social cues to affective changes, prompting upregulation processes? This is only one of many possible mechanisms which, if detected, could be added to the athlete’s and practitioner’s arsenal. A good start point would be to consider the overlaps between the strategies proposed to underpin PT switching in MAP and the impact of the different sources of self-efficacy.

### 3.2. Perception of Control (Previously Examined in MAP)

In contrast to self-efficacy, perceived control was examined in a number of MAP studies [4,7]. In its current format, perceived control is a fundamental aspect that differentiates between and characterizes the four PTs. Specifically, Bortoli and colleagues characterize Type 1 as automatic performance, requiring the athlete to supervise performance; Type 2 is characterized as optimally controlled, with an effective recruitment of resources; Type 3 as over-controlled, resulting in reinvestment; and Type 4 as under-controlled, and lacking in focus [1].

In relation to the MAP typologies, perception of control was investigated both prior and following performance. For example, in terms of pre-performance, performers’ perceived control over idiosyncratic core components was examined [7,13]. Post-performance, participant self-rated levels of perceived control post-shot were gathered, before objective shooting scores were revealed to them [4,8,9]. Moreover, self-report scores of perceived control were utilized to categorize PTs for each participant [8,9,13], reflecting the 2 × 2 interaction between performance experience and level of control.

Further to assisting with the categorization of PT’s, the investigation of perceived control could also provide valuable insight into when, and why, an athlete may move between PTs. Specifically, perceived control could act as a cue for the need to transition. With regard to Type 1, meta-analytic findings on controllable and uncontrollable elements of flow state corroborated previous findings [37,38]. This research concluded that skills to maintain flow were at least partially controllable; this was in contrast to disruptors of flow, which were deemed uncontrollable by participants [37]. Therefore, it could be valuable to gain qualitative insight into the role of perceived control over potential disruptors of Type 1 performance, coupled with effective coping strategies that were employed to maintain a PT.

Regarding Type 2, MAP proposes that effort is consciously exerted, although affect is negative [1,4]. This is particularly interesting against the backdrop of Wood and Wilson’s findings [39]; they established that high self-ratings regarding perception of control beliefs resulted in better overall performance, less anxiety about performing well, and reduced levels of perceived uncertainty. There are clear parallels between these findings and those for Type 2 [39], except that the MAP framework associates negative affect with Type 2. Consequently, future research is required to investigate Type 2 performance more closely, specifically, the role of affect and emotion, as Type 2 could be even more multi-faceted and dynamic than currently assumed.

Regarding suboptimal performance, low perceived control, along with distraction and debilitative anxiety, may be responsible for suboptimal performance [40]. This may be specific to Type 3 performance, where it is currently proposed that conscious control of automated movement patterns through declarative knowledge disrupts performance [19,40]. This negative interaction between cognition and the attendant movement patterns highlights contrasting dimensions of control: while an athlete may experience low perceived control as a result of a stressor, one ineffective coping strategy could be to exert too much attentional resource and, therefore, over-control and underperform. This complex dynamic is exacerbated by negative affect leading to a further decrease in perceived control and the application of less effective coping behaviors (e.g., [41,42]). Future research is needed to corroborate current hypotheses on coping strategies [1], examining the role of perceived control in upregulating from suboptimal to optimal performance, as well as the efficacy of coping behaviors such as cognitive restructuring [43].

### 3.3. Implications: Self-Efficacy and Perceived Control as Cues for Transition Processes

Based on the review above, it becomes clear that both self-efficacy beliefs and perception of control are crucial for a better understanding of transitions between PTs. They present critical *cues* for the need to transition and upregulate when negotiating stressors, while acting as *precursors* to employing coping strategies and behaviors. Future research is needed to consider this interesting dynamic more closely. For example, could high self-efficacy compensate for low perceived control for athletes operationalizing Type 3 performance? Or, which associated coping strategies are most effective in upregulating to optimal performance? Furthermore, in the context of self-efficacy, an examination of the links between appraisal (e.g., available coping resources versus required coping efforts) and perceived control beliefs could provide valuable insight into the nature and dynamic of Type 2 performance.

In terms of specific tools used to synthesize and apply these findings, it would be instructive to investigate the potency of self-talk, specifically to help athletes transition between PTs as they are contextualized in MAP. For example, what types of self-talk are best suited for what types of transitions, and how can practitioners train athletes in their effective application to recognize specific PTs and employ potential solutions, accordingly? Future research needs to address these questions with consideration given to existing findings regarding the context-dependent nuances of self-talk in high performance, i.e., organic vs. strategic and spontaneous vs. goal-directed [44,45].

Overall, we would suggest that MAP research, at least so far, underutilized perception of control as a variable. While useful for categorizing PTs idiosyncratically, perceived control, along with self-efficacy beliefs, could be used to better understand athletes’ evaluation of their internal experience, task, and context in relation to PTs, as well as movement between them. Indeed, the current understanding of peak performance, as delivered by MAP, could be extended beyond emphasizing what happens during PTs, towards examining mechanisms of upregulation and coping in athletes, which would carry critical implications for applied practitioners.

## 4. Post-Transition: Effort and Reward Perception

To date, MAP research elicited insightful, multi-faceted findings that uncovered what athletes think, feel, and experience at both a psychological and physiological level during peak and non-peak performance states. Moreover, MAP research covered a range of experimental timeframes, ranging from a single experimental session [4,7,8,9] and multiple lab visits [10,11,12,13], to longitudinal research [1]. One aspect still requiring examination within MAP, however, is the short- and long-term impact of operationalizing a given PT on the athlete’s ability to upregulate. For example, what is the reward perception, as experienced by the performer, of executing seemingly automatic Type 1 performance, in comparison to effortful Type 2 performance? Rewards exert the greatest effects in anticipation, as opposed to actual achievement, and manifest in a dopamine response [46]. Consequently, gaining insight into athletes’ reward perception, especially in terms of their short- and long-term expectations, could be useful for better understanding why athletes (re)operationalize a specific PT in a corresponding, albeit idiosyncratic, performance context.

### 4.1. Intrinsic Motivation in Type 1 and 2

Intrinsic motivation, as conceptualized within self-determination theory (SDT), is characterized as a distinct, inherent satisfaction with task execution alone [47,48]. In research on flow states, such inherent satisfaction is termed “autotelic” [18,22]. Moreover, extrinsic, tangible rewards were found to decrease intrinsic motivation [49]. However, intrinsic motivation increases with heightened perceived control, which is, within MAP, at its optimal level in Type 1 and Type 2 performance [4,46]. Moreover, intrinsic motivation, inherent in autotelic satisfaction, was established as the reward structure behind flow experience and, therefore, Type 1 performance [22]. Indeed, during flow, performers engage in task-relevant, positive emotions, which, coupled with intrinsic motivations, makes performers want to reacquire this state [22]. Swann and colleagues extended this framework by highlighting that, following successful performance, both flow and clutch states reward the performer intrinsically [26]. Importantly, however, from the performer’s perspective there are appreciably different emotional experiences resulting from flow states versus a clutch performance experience. While the performer may feel energized after a flow state experience, clutch performance usually leaves them emotionally and physically drained [26]. In the context of MAP, both Type 2, and especially Type 3 PTs leave the performer fatigued. Transferring these findings to the SDT framework, one could summarize that while Type 1 is inherently intrinsically motivating, it takes effort for the athlete to *(self-)determine* performance, whereas in Type 3, the athlete’s performance is *determined* by challenges and obstacles. Future research is needed to explore this dynamic and examine sources of intrinsic motivation leading to continued engagement in performance.

### 4.2. Perception of Effort in the MAP Framework

Within the MAP framework, it was established that both Type 1 and Type 4 involve less effort, in contrast to Types 2 and 3, where athletes exert significant effort [1]. Effort can be categorized into initial effort mobilization and subsequent effort to maintain, enter, and exit a PT. Effort to *maintain* a PT was examined through measuring participants’ ratings of perceived exhaustion associated with their idiosyncratic PT profiles [11]. However, the (mental and psychological) effort required to *transition* between PTs is yet to be investigated within MAP. For example, in Type 4, where athletes perform sub-optimally with little conscious control, effort mobilization may in part depend on a performer’s intensity of motivation and positive affect in order to transition to a more favorable PT (Motivational Intensity Theory [50,51]).

Regarding Type 2, MAP’s current association of negative affect with high effort presents a contrast to existing literature. Acknowledging that the performer deals with stressors in this PT, how can something that is optimally controlled and produces optimal performance result in negative affect? For example, higher levels of effort were found to correspond to an increased level of perceived value, if performance was already successful [52]. Further, individuals generally expect larger rewards when greater effort had to be exerted to perform a task [53]. Within MAP, this could mean that given the efforts involved in transitioning into and maintaining Type 2 performance, Type 2 could be, once the performance is complete, perceived as highly rewarding to the athlete.

Considering that optimal performance itself may be—if intrinsic motivation is the driving force—the reward behind Type 2 performance, these findings suggest that the social environment that corresponds to Type 2 performance requires further investigation. For example, if an athlete exerts conscious effort to produce optimal performance, but recognizes that their current experience is highly efficacious, this should provide them with a positive emotional experience that motivates them to engage and subsequently re-engage with this process. As such, perceptions of self-reward may offer an additional dimension and enable the differences highlighted across “making it happen”/Type 2 to be teased out.

### 4.3. Implications: Reward Perception as an Idiosyncratic Facet of Performance

As referred to previously, flow is elusive, complex to attain, and easily disrupted [22,54]. In contrast, Type 2 presents optimally controlled, effortful, superior performance [1]. This poses the question whether, once performance is completed, Type 1 performance is accompanied by the same feeling regarding accomplishment, compared to Type 2. For example, future research needs to examine which PT an athlete is more likely to re-engage with, given their perception regarding attainment, execution, and task completion? Furthermore, the reality for most elite/pre-elite performers is that they operate close to their performance ceiling, this being an important precursor of attaining and maintaining their level. Consequently, it is important to determine whether there is a difference between short-term and long-term reward perception and how this affects MAP typology. To gain insight into these questions, longitudinal research is needed to investigate biopsychosocial differences between athletes who exhibit a greater propensity to operationalize one PT over the other(s).

Moreover, regarding Type 1, flow state may require optimal, post-voluntary levels of effort, opposed to being a purely effortless state [55]. Future research is needed to investigate the precursors of flow that allow performers to enter what they would term a ‘flow’ state. Consequently, it is key to determine whether cognitive, emotional, and behavioral antecedents can be reproduced. In addition, research is needed to investigate the role of effort in this dynamic. Regarding Type 2, future research may examine the role of affect and emotions in Type 2, as well as their connection to situational appraisal and self-regulation. For example, is there a dimension of Type 2 performance, characterized by a challenge appraisal and positive affect? Overall, reward value depends on the goals of an individual [56]. It can be assumed, that in Types 1 and 2, an individual’s aim is to maintain optimal performance, whereas Types 3 and 4 are characterized by self-regulation and coping to attain optimal performance. However, future research is needed to examine this dynamic.

## 5. Discussion

The purpose of the present review was to synthesize research conducted within the MAP framework to highlight both our current understanding of MAP and to identify potential gaps and avenues for future research. Specifically, we made a case for examining transitions between mental states, and their associated PTs. Subsequently, we presented our ideas on investigating pre-transition psychosocial cues and post-transition effort and reward perception. We demonstrated that all three areas could provide valuable insights into when, why, and potentially how athletes transition between PTs. This would enhance MAP and enable the model to depict transitions from suboptimal to optimal performance, as well as provide insights as to the maintenance and continuance of Types 1 and 2.

### 5.1. MAP’s Contributions to Date

MAP was designed as a multi-measure intervention model for sport performance. It successfully marries mechanistic, psychophysiological evidence with evidence of behavior, cognition, emotion, attentional focus, and performance measures, establishing itself as a robust model and informing guidance for practitioners. A variety of experimental timeframes (i.e., longitudinal, cross-sectional) and contexts (i.e., practice sessions, performance simulation, physiology labs) are employed, each specifically tailored to realize research aims and methodologies. Furthermore, MAP approaches performance as an individualized, idiosyncratic concept; consequently, this results in specific findings, relevant to practitioners, athletes, and coaches. Studies were creatively constructed specific to performance contexts and involved expert performers, which resulted in the creation of high-quality data sets. The consistent categorization of PTs led to a coherent body of research, acting as a solid foundation for future investigations. In conclusion, MAP can be characterized as a “quality over quantity” model. Importantly, however, as in any scientific work, there is potential to expand its scope and learn more about its applicability and utility to performers.

### 5.2. Essential Next Steps: Directions for Future Research

As discussed earlier, the MAP model was empirically investigated in closed-skill, self-paced activities (e.g., [7]), endurance sport (e.g., [10]), and driving simulation [13]. As a next step in the development of MAP, we propose the examination of PT characteristics in an open-skill, externally paced, hyper-dynamic performance environment—for example combat sports. Such efforts would reflect the variety of biopsychosocial demands on the performer (depending on the type of activity), and make MAP into a more comprehensive, applied, and accessible model for practitioners and coaches.

Second, MAP research to date focused on investigating what happens during operationalization of PTs (i.e., characteristics and concomitants), but not on how athletes can best transition between PTs. Future research needs to carefully consider task, context, and impact, using qualitative and quantitative evidence to better understand the processes that enable transition between PTs, both on a micro, e.g., during performance, as well as on a macro level, e.g., throughout a season. This could be explored in conjunction with contextualizing clutch performance within a MAP framework since the clutch can occur episodically during performance [25].

Further, future research is required to examine how transitions within MAP progress. For example, is there a linear progression between stages (e.g., 4, 3, 2, 1 and vice-versa)? Can PTs be skipped (e.g., from Type 3 to Type 1)? Research indicated that emotion-focused coping strategies can aid transitions from Type 3 to Type 1, and that action-focused coping strategies facilitate transitions from Type 3 to Type 2 [1]. However, this is yet to be observed and documented through empirical study.

Moreover, it is still unclear what cognitive, emotional, or environmental cues initiate PTs. Can a performer train to function in a Type 2 state, peaking, occasionally/unconsciously, into Type 1? Furthermore, could trait affect and/or subjective, context-specific emotions act as meta-cognitive cues for initiating corresponding PTs (see Affect as Information Theory [21])? These findings would not only expand our current theoretical understanding of self-regulation and coping, but also enrich practitioner and coaching literature in terms of how athletes can be best supported in attaining, and maintaining, optimal performance.

### 5.3. Essential Next Steps: Practical Implications

In an applied setting, the MAP model could also be incorporated into existing intervention methods to prevent or manage “choking” under pressure [57]. In addition, MAP findings could offer applied guidance for performance refinement and for preventing underperformance such as in Types 3 and 4 [9,58]. Further, it could be insightful to take stock of the current application of the MAP framework in practice. As academics *and* practitioners, we not only inform practice through research, but also hope to gain insight from those applying the research, to ask relevant and specific research questions. Reflecting this, future research could explore how practitioners, coaches, and athletes utilized MAP research thus far, as well as gaps they would like to see addressed.

## 6. Conclusions

The employment of a mixed methods approach in relation to MAP yielded important and insightful findings to date, and it is important these modes of research continue. Gaining access to rich data regarding athlete experience, coupled with quantitative methods examining psychophysiological underpinnings of performance, holds further, significant promise.

Currently, MAP treats PTs as states; however, considering that transition processes between PTs might be dynamic and complex, it could be valuable to study the underlying contextual and individual differences that will likely inform the MAP typology that may fit a performer’s particular mindset. In short, entering a MAP typology may be more involved than “just” a switch, as is currently assumed [1].

Lastly, in part due to its conceptual heritage, MAP is currently presented as a sport-specific model. However, aspects such as dealing with stressors, flow state, effort exertion, as well as the contrast in markers of optimal, in comparison to suboptimal performance, are highly transferable to other domains of human performance. The idiosyncratic nature of MAP research findings is a methodological strength and can yield further valuable insights across performance domains, coupled with an in-depth understanding of performance context and culture.

## Data Availability

Not applicable.

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
