# Peer review of "Reading the MAP: A Pracademic Perspective on the Current State of Play of the Multi-Action Plan Model with Regard to Transitions between Mental States"

_ijerph, 2022, doi:10.3390/ijerph192315520_

Round 1

Reviewer 1 Report

I appreciate the time the authors took to write this manuscript, where a narrative synthesis was used to illustrate the roles of psychological factors. The literature review provided a satisfactory overview of different mental states, what is known about the subject area. However, the introduction of the literature on the moderators and its relationships would be welcome. With that said, there are some concerns at present that I encourage the authors to explain how and when the proposed framework can occur to examine the moderating effects between two or across the performance types. Further, there were crucial elements missing from the methodological point of view. For example, taking a somewhat systematic review/meta-synthesis could make a robust integration of the findings from existing studies and do much with this approach to propose a strong framework. I explain my concerns in more detail below and hope the authors could use some to strengthen the manuscript.

Intro

I found the intro was structured and articulated clearly but could go further to nail the key points and examples to provide some interesting discussion and bring ideas together more coherently. For instance, a brief intro and comparison of existing frameworks mentioned in the 3rd para (lines 54-65, which is anyway quite short) would set out an area to which the moderating effects can contribute to the current literature and MAP model.

Page 3, lines 123 and onwards, the authors presented a good overview of the model and relevant research focused on the noncontact, unorganised sport setting. Elaboration of each dimension in a specific sport domain or movement could be worthy and appealing, setting the scene of the moderation framework (see page 4, line 191-192 for the same concern).

Page 3, line 145 and onwards, some descriptions and critical discussion on choking/clutch performance would communicate key findings from extant research more clearly, which may highlight vital differences in the performance types 1 and 2. Add citations to back up the argument on line 162.

LR

Page 4, line 177 and onwards, expand more to take shape, a distinct argument concerning how the current work is different from the work by Bortoli et al. (2012) and elicits additional meaning to the literature presented earlier. Perhaps some limitations in the previous studies might help signpost and establish how the moderating factors can contribute to the literature.

Page 5, lines 223-235, please provide evidence to back up the argument made.

Page 6, lines 274-286, went off on a tangent when incorporated two unrelated ideas. Ranges of sources are required to support this gap in the literature.

The major issues in the sections 2 and 3 were there was, in places, little engagement with literature to synthesise and support the findings. The propositions were allusive rather than rigorous review/arguments on the relationships between individual-level factors and the four types of performance to frame the current study. e.g., lines 360-367, is not there any existing research covering how these self-talk techniques are associated with different types of performance? What about the three self-talk techniques? was this discussion relevant with the scope of the study?

Finally, the current study was with no rigorous search strategy identified and no justifications for data selection. Why did the authors review and integrate existing studies and in what criteria? In this regard, I think there was scope for further development to add depth and methodological rigour by using a systematic review or meta-synthesis.

Editorial elsewhere

Page 1, line 19 and elsewhere, consider a different, specific term rather than this vague, allusive one (i.e., biopsychosocial concept) since the details of each dimension were not fully addressed in the manuscript.

Page 1, line 34, remove the last comma before “sits.”

Page 2, line 49, S/V agreement = are.

Page 2, lines 45-81, too many ideas in a para. Break into the two/multiple paras to flow better.

Page 3, line 121, can it be in the reversed order? e.g., 4-1 to 1-4.

Page 3, line 132, remove the comma before “that.”

Page 4, lines 157-159, somewhat fragmented and incomplete.

Page 4, line 174 and onwards, define timescales to refer to a short- and long-time frame, particularly in the current study (see page 5, line 224 for the same concern. What is meant by micro, meso, and macro performance episodes?).

Page 5, line 243, the authors took a slightly confused approach to the subheading and terms. e.g., given the review (and that self-efficacy, perceived control, perceived reward, etc. are individual-level factors), socio-environmental precursors mentioned were quite inappropriate as a running subheader. The use of psychological factors could be more communicative and less ambitious, shying away from big words.

Page 5, line 206, remove “s” from events.

Page 6, lines 277-279, somewhat awkward. Make a complete sentence.

Page 6, line 279, “What PTs do.”

Page 7, line 348, consider replacing “findings” with “review” as it is read as the findings from the study.

Page 10, lines 506-509, lacking clarity. Was it about action-focused coping strategies?

Page 11, line 524, define what it means by pracademics.

Reviewer 2 Report

Thanks for the opportunity to review this paper. This study present a synthesis of current understanding of MAP research, coupled with offering applied implications and directions for future research

This manuscript have a significant contribution to the field, well organized and comprehensively described, scientifically sound, and there appropriate and adequate references to related and previous work, contains original and significant information that justify its publication in the International Journal of Environmental Research and Public Health.

However, below I present reflections in order to improve some issues addressed in the text.

First, author argue exhaustive each PT operates on an affective, cognitive, behavioral. I miss out psychophysiological level in the manuscript. Could you write more about this performance component in each PT?

Second, could you write about difference between self-confidence and self-efficacy in each PT?

I understand that self-confidence is a belief and self-efficacy is a judgment about ability to execute a specific task effectively.

Third, I suggest that authors rewrite the intrinsic motivation in Type 1 and 2. Could you write about the self-determination theory instead intrinsic motivation?

Fourth, I suggest two specific topics:

a)      Practical implications. How athletes, coaches, coach staff could use the MAP?

b)      How development experimental/ clinical/ intervention studies about MAP?
